# Integrated Biorefinery of Empty Fruit Bunch from Palm Oil Industries to Produce Valuable Biochemicals

**Rendra Hakim Hafyan [1], Lupete K. Bhullar [2], Shuhaimi Mahadzir [1,*], Muhammad Roil Bilad [1] , Nik Abdul Hadi Nordin [1], Mohd Dzul Hakim Wirzal [1] , Zulfan Adi Putra [2] , Gade Pandu Rangaiah [3] and Bawadi Abdullah [1,4]**

1   Department of Chemical Engineering, Universiti Teknologi PETRONAS, Bandar Seri Iskandar 32610, Malaysia; rendra_17006879@utp.edu.my (R.H.H.); mroil.bilad@utp.edu.my (M.R.B.); nahadi.sapiaa@utp.edu.my (N.A.H.N.); mdzulhakim.wirzal@utp.edu.my (M.D.H.W.); bawadi_abdullah@utp.edu.my (B.A.)
2   PETRONAS Group Technical Solutions, Process Simulation and Optimization, Level 16, Tower 3, Kuala Lumpur Convention Center, Kuala Lumpur 50088, Malaysia; lupetebhullar@gmail.com (L.K.B.); zadiputra123@gmail.com (Z.A.P.)
3   Department of Chemical and Biomolecular Engineering, National University of Singapore, Singapore 117585, Singapore; chegpr@nus.edu.sg
4   Chemical Engineering Department, Center of Contaminant Control and Utilization (CenCoU), Institute Contaminant Management for Oil and Gas, Bandar Seri Iskandar 32610, Malaysia
*   Correspondence: shuham@utp.edu.my; Tel.: +60-5-368-7622

**Abstract:** Empty fruit bunch (EFB) utilization to produce valuable bio-chemicals is seen as an economical and sustainable alternative to waste management in palm oil industries. This work proposed an integrated biorefinery configuration of EFB valorization considering sustainability pillars—namely, economic, environmental, and safety criteria. Techno-economic analysis, life cycle assessment, and hazard identification ranking methods were used to estimate annual profit, global warming potential (GWP), fire explosion damage index (FEDI), and toxicity damage index (TDI) of the proposed integrated biorefinery. A multi-objective optimization problem was then formulated and solved for simultaneous maximization of profit and minimization of GWP, FEDI and TDI. The resulting Pareto-optimal solutions convey the trade-off among the economic, environmental, and safety performances. To choose one of these optimal solutions for implementation, a combined approach of fuzzy analytical hierarchy process and a technique for order preference by similarity to ideal solution was applied. For this selection, the economic criterion was more preferred, followed by the safety and environmental criterion; thus, the optimal solution selected for integrated biorefinery configuration had the highest annual profit, which was at the maximum capacity of 100 ton/h of EFB. It can fulfill the global demand of xylitol (by 55%), levulinic acid (by 98%), succinic acid (by 25%), guaiacol (by 90%), and vanillin (by 12%), and has annual profit, GWP, FEDI, and TDI of 932 M USD/year, 284 tonCO$_2$-eq, 595, and 957, respectively.

**Keywords:** empty fruit bunch; palm oil industry; integrated biorefinery; techno-economic analysis; life cycle assessment; inherent safety; multi-objective optimization; fuzzy analytical hierarchy process; technique for order preference by similarity to ideal solution

## 1. Introduction

Today, the issue of sustainability has been a major concern associated with palm oil production. The palm oil industry has brought a positive impact on the economic growth in Malaysia since it created jobs as well as downstream activities for national development [1]. It has been the major

contributor from the agriculture sector by 8% or over RM 80 billion of the gross national income [2]. Nevertheless, some negative impacts caused by palm oil production have led to criticisms such as deforestation, land-use change, waste generation, greenhouse gas emissions, etc. [3]. Furthermore, only 10% of palm oil trees can be converted into palm oil products [4]. The remaining 90% of palm oil trees generate massive biomass residues creating environmental issues [5], despite having the potential to be co-processed in the petroleum refinery [6]. One of the abundant residues is empty fruit bunch (EFB), which is a waste product after the extraction of oil from the fresh fruit bunches (FFB). An estimate reveals that the EFB produced is 22% of the palm oil [7,8]. However, on the positive note, EFB is a potential source to be converted to value-added biochemicals due to its high content of cellulose and hemicellulose [9]. In the current practice, waste management of EFB use from palm oil as a potential organic manure for agricultural purposes had been conducted to deal with this environmental issue [10–16].

Efforts have been made to reduce palm oil waste by the Malaysian government through various strategies, such as renewable energy policy, the National Biomass Strategy 2020, and the 1 Malaysia Biomass Alternative Strategy (1MBAS) to encourage the utilization of biomass for value-added products and bioenergy production [17,18]. Previous studies had tried to improve the sustainability of the palm oil industry by valorizing the palm oil waste into several products. Yoshizaki et al. [19] proposed a new approach for the integrated technology of biogas energy and compost production for a palm oil mill. The integrated biogas compost technology was found to be the most economically viable alternative for biomass utilization at the mill. The study by Ali et al. [20], which extended the work of Yoshizaki et al. [19], proposed a sustainable and integrated biorefinery concept for the palm oil mill, which showed a huge potential to enhance the economic improvement locally by generating a potential profit of RM 5.6 million and 15–20 new jobs at any given mill as well as reducing emissions. Kasivisvanathan et al. [21] examined an integrated palm oil-based biorefinery by incorporating multiple biomass processing platforms with combined heat and power (CHP). In this case, EFB, Palm Kernel Shell (PKS), Palm Oil Mill Effluent (POME), and Mesocarp Fiber (MF) were identified as the potential biomass feedstock. These biomasses were further processed to produce valuable products (i.e., dried long fiber), pellet, briquette, charcoal, compost, etc.), and energy generated from the given palm-based biomass in CHP was integrated with palm oil-based biorefinery.

Tan et al. [22] presented an integrated palm oil complex concept to utilize POME. Their study involved the economic, environmental, and social concerns to review the favorability of POME utilization. Three different cases in the POME utilization are biogas-biomass co-fired boiler, on-grid biogas plant, and POME evaporation. Foong et al. [23] developed an input–output optimization approach for sustainable palm oil plantations. Their model involves determining the best management practices through optimal fertilizer application to maximize the yield of palm oil, thus minimizing the need for further land expansion. Aristizabal et al. [24] presented an integrated palm oil biorefinery to produce different bioenergy products such as biodiesel, bioethanol, butanol, hydrogen, and electricity. Kasivisvanathan et al. [25] developed a flexibility model to deal with the uncertain product demand towards the capacities of palm oil-based integrated biorefinery. In their work, palm oil waste was utilized to produce various bio-based products—namely, acetone, animal feed, biochar, biofuel, bio-oil, butanol, steam, and ethanol.

An interesting palm oil waste conversion to produce high-value marketable biochemicals in the form of integrated biorefinery has not been explored in the previous works. According to the International Energy Agency (IEA) Bioenergy Task 42, the concept of a biorefinery has gained critical attention to provide a sustainable process technology of biomass to a wide range of profitable chemical products; biorefineries are also seen as a sustainable waste management that can promote economic opportunities, environmental benefits, and energy security [26]. Isikgor et al. [27] identified around 200 potential valuable biochemicals and biopolymers derived from biomass. The department of energy (DOE) had identified the top 12 chemicals from biomass, primarily based on their market potential [28]. Sadhukhan et al. [29] revealed that the production of fine, specialty and platform

chemicals, polymers, food and pharmaceutical ingredients, alongside biofuel and bioenergy, from palm oil waste can attain overall sustainability by the replacement of fossil resources. A wide movement to shift from non-renewable to renewable and sustainable resources to produce biochemicals is expected to reduce the dependence of fossil-based resources as well as mitigate the climate change issue [30]. Furthermore, fluctuating oil prices and consumers' awareness in using eco-friendly products have opened up opportunities for bio-based chemicals to replace fossil-based chemicals. Also, manufacturers see bio-based chemicals as economically attractive since they have great potential to develop bio-based building block chemicals and specialty chemicals [31].

In the development of palm oil-based biorefinery and its sustainability aspects, there have been several works in the past. Adulrazik et al. [32] designed an optimal EFB supply chain for the multi-products production of energy and chemicals considering economic aspects. Lim et al. [3] assessed the sustainability of the crude palm oil supply chain in Malaysia, taking into account economic, environmental, and social aspects. Following Abdulrazik et al. [32], Rubinsin et al. [33] developed an optimal supply chain that integrated availability, locations, processing technologies, capacities, bioproducts, and power production by simultaneously considering economic and environmental aspects as an expansion approach for the palm oil industry. Kasivisvanathan et al. [21] adapted fuzzy optimization for designing the retrofitted palm oil-based integrated biorefinery considering economic performance and environmental impact. Andiappan et al. [34] presented an integrated approach of the palm oil industry by determining the optimal allocation of incremental profits and corresponding sustainability and economic viability. Munasinghe et al. [35] addressed the optimal planning of the supply chain of crude palm oil production considering economic, environmental, and social aspects. Vaskan et al. [36] presented an economic and environmental impact evaluation for the valorization of EFB to produce bioethanol, C5 syrup as cattle feed, power, and heat. Foo et al. [37] proposed a linear programming (LP) model to address the optimum allocation of palm oil biomass taking into account profit maximization and cost/$CO_2$ minimization. Foong et al. [38] addressed the optimal palm oil milling process considering economic evaluation using a hybrid combined mathematical programming and graphical approach. Tan et al. [39] proposed an LP cooperative game model to allocate benefits that accrue from inter-plant integration in an eco-industrial park (EIP).

In this study, the abundant and underutilized EFB has been selected as a feedstock to produce various high value chemical building blocks, which have not been considered in the previous papers. Bhullar [40] had identified sugars-based and lignin-based chemicals that have high economic potential; they are xylitol, levulinic acid, succinic acid, guaiacol, and vanillin, which are considered in this work. The growing concern of sustainability has increased the importance of taking economic, environment, and safety aspects into consideration as the criteria for the integrated sustainable biorefinery. Yet, most of the previous studies on palm oil-based biorefinery have focused on economic and environmental performance, and/or limited to individual chemicals such as levulinic and succinic acids [41,42]. Safety assessment is infrequently conducted at the early design stage, and it is commonly performed in the detailed design phase due to the nature of process safety methodologies, e.g., HAZOP. It is expected that, by simultaneously incorporating economic, environmental, and safety aspects, the conceptual design of an integrated biorefinery will lead to sustainable and optimal processes and consequently biorefinery establishment, as demonstrated for other cases in our previous works [43–45].

In this work, annual profit is estimated by Techno-Economic Analysis (TEA), Global Warming Potential (GWP) is quantified using Life Cycle Assessment (LCA), and Fire and Explosion Damage Index (FEDI) and Toxicity Damage Index (TDI), categorized as hazard potential, are performed using Hazard Identification and Ranking Assessment (HIRA) from Khan et al. [46]. The goal is to find an optimal EFB-based biorefinery configuration, simultaneously considering economic, environmental, and safety aspects. The design problem is presented as a multi-objective optimization (MOO) problem, which is solved by a multi-objective genetic algorithm to determine optimal capacities, yield and biomass allocation lead to maximum profit, the minimum of both global warming potential and hazard level. The obtained results (the so-called Pareto-optimal solutions) reveal the trade-off among the

objectives. In the subsequent step, a combined approach of Fuzzy Analytical Hierarchy Process (FAHP) and Technique for Order Preference by Similarity to Ideal Solution (TOPSIS) is applied to select one of the Pareto-optimal solutions.

## 2. Methodology

The proposed integrated biorefinery model is depicted in Figure 1, where EFB generated from palm oil mill activity is collected and sent to a centralized integrated biorefinery area. The potential of converting other palm oil wastes to valuable products is not considered in this work. A flowchart of the methodology of the present study is outlined in Figure 2. EFB composition is taken from Chiesa et al. [47]. Dilute acid pretreatment, enzyme production, and saccharification processes are selected and simulated in Aspen Plus V10 according to Humbird et al. [48]. The number of conversion steps for this work is limited to five biochemical products according to the potential revenues, and they are xylitol [49,50], levulinic acid [51], succinic acid [52], guaiacol [53], and vanillin [54]. The process simulation results are then used for equipment sizing and sustainability assessment.

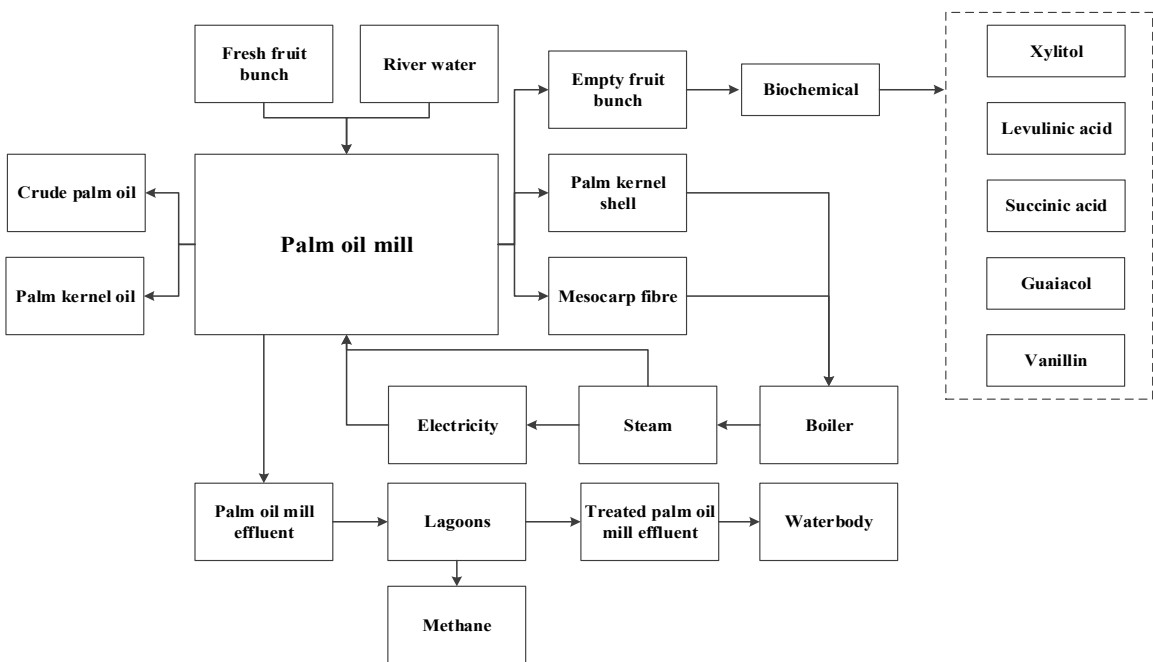

**Figure 1.** Palm-oil-based integrated biorefinery concept [55].

For sustainability assessment of the integrated biorefinery, TEA is used to estimate the annual profit, LCA to quantify GWP, and HIRA to measure the hazard potential at initial design in terms of FEDI and TDI. These multiple criteria lead to a MOO problem to simultaneously maximize annual profit and minimize GWP, FEDI, and HIRA. The decision variables involved are biomass availability and product yields, and the constraints are related to mass and energy balances and global products demand. Prior to solving MOO problems, the developed process simulation models are simplified as surrogate models relating decision variables with the objectives [56]. The surrogate (regression) models are developed from rigorous simulation results obtained through a central composite design (CCD) for later use in solving the MOO problem. The solution of MOO problems produces Pareto-optimal solutions, which reveal the trade-off among the objectives. To choose one of these solutions, FAHP and TOPSIS are used; FAHP takes into account the subjective inputs of experts with profound knowledge [57].

The proposed superstructure for biorefinery based on EFB is shown in Figure 3. The underutilized EFB supply is collected and transported to a central processing facility. Transportation costs and supply chain issues are not considered in this work. The EFB then undergoes a pre-treatment process to break down its major components into cellulose, hemicellulose, and lignin, which will be used

as the feedstock to subsequent processes to produce xylitol, levulinic acid, succinic acid, guaiacol, and vanillin.

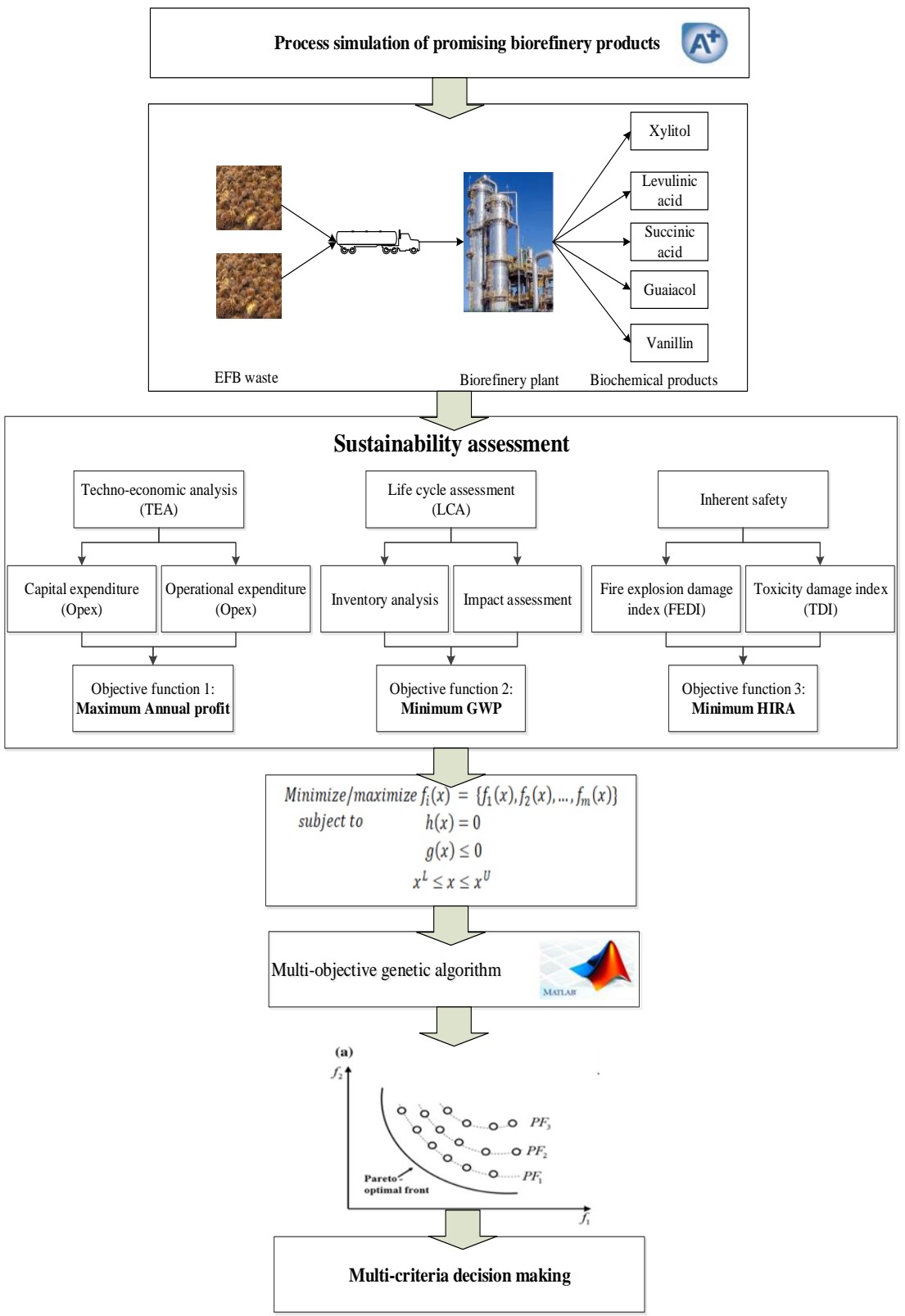

**Figure 2.** Methodology for identification of optimal empty fruit bunch (EFB)-based biorefinery configuration.

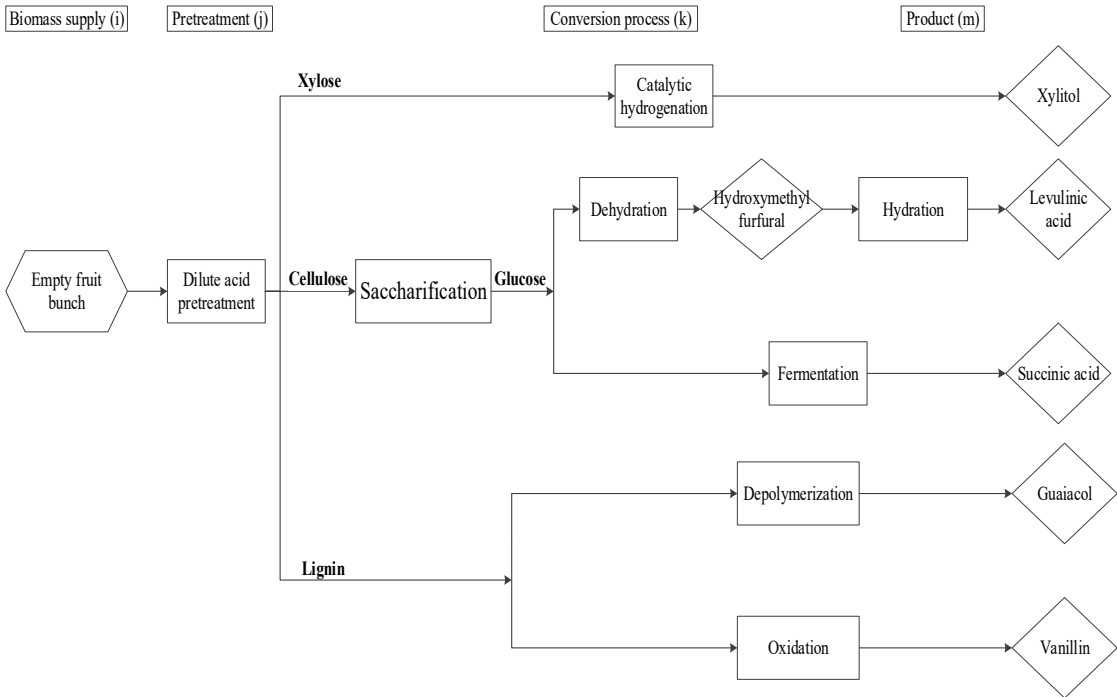

**Figure 3.** Schematic of the integrated biorefinery from the empty fruit bunch (EFB).

### 2.1. Techno-Economic Analysys

TEA of the processes is estimated using Aspen Process Economic Analyzer (APEA) V.10. The basis for this estimation is the simulated results and equipment sizing. Investment factors to calculate project capital expenditure are taken from Peters and Timmerhaus [58], as tabulated in Table S1 in the Supplementary Materials. The estimation of annual operating cost is based on Turton et al. [59], and details of this estimations are reported in Table S2 in the Supplementary Materials. The annual profit is calculated as the difference between the product annual sales and total cost, which is the sum of annualized capital expenditure (CAPEX) and annual operating expenditure (OPEX). The economic objective is to maximize annual profit ($/year), as shown in Equations (1)–(3).

$$profit = \sum_{m} B_m^{prod} C_m^{prod} - \sum_{j} \sum_{k} Ca_{jk} - \sum_{j} \sum_{k} Op_{jk} \tag{1}$$

$$Ca = \sum_{j} \sum_{k} Tdc + Tidc + Cfe + Con + Fci \tag{2}$$

$$Op = \sum_{j} \sum_{k} Dmc_{st} + Fmc_{st} + Gmc_{st} \tag{3}$$

Here, $B_m^{prod}$ (kg/h) is the mass production rate of products (xylitol, levulinic acid, succinic acid, guaiacol, and vanillin) and $C_m^{prod}$ ($/kg) is the product price. $Ca_{jk}$ ($/year) is annualized CAPEX of the pretreatment and final processes. The CAPEX is the sum of total direct plant cost, total indirect plant cost, contractor's fee, contingency and fixed capital investment. $Op_{jk}$ ($/year) is equal to sum of direct manufacturing cost, fixed manufacturing cost and general manufacturing cost.

### 2.2. Life Cycle Assessment

The environmental impact assessment of the integrated biorefinery adopts the LCA technique using the ISO standard [60]. All emissions are taken from the developed process simulation such as direct emissions of the process, electricity and heat consumption. The emission data related to

electricity and heat consumption are acquired from commercial LCA databases such as Ecoinvent 3.5 [61] and Simapro 8.5.3 [62]. Then, the total Life Cycle Inventory (LCI) can be formulated as a function of the direct emission, electricity and heat consumption, as stated in Equation (4).

$$LCI_{tot} = LCI^{process} + LCI^{electricity} + LCI^{heat} \tag{4}$$

The *LCI* is then translated into the corresponding environmental impact (GWP). The emissions considered are carbon dioxide ($CO_2$), methane ($CH_4$), and nitrous oxide ($N_2O$). The GWP is calculated as the total of GWP from each of these emissions as shown in Equations (5) and (6).

$$GWP = \sum_i LCI_{tot} \times m_i \tag{5}$$

$$GWP_{tot} = \sum_j GWP_j + \sum_k GWP_k \tag{6}$$

In this equation, $m_i$ is the damage factor that accounts for each greenhouse emission, which is retrieved from Guinee [63]. $GWP_j$ denotes the GWP in the pretreatment process while $GWP_k$ indicates that of the chemical production process.

## 2.3. Inherent Safety

Two hazards potential, quantified in this work, are FEDI and TDI [46]. FEDI is based on thermodynamic data, where the energy factor is the main aspect of the calculation. Another aspect is penalty, which is given based on the operating range of the process. The penalties involve five categories—namely, storage units, units involving physical operations, units involving chemical reactions, transportation units, and other hazardous units. Finally, the damage potential of FEDI is found by the multiplication of the penalties and the energy factors. For TDI calculation, toxic load contained in the processing unit is the major factor. Physical and chemical reaction units are considered. Penalties typically assigned for the location of the nearest hazardous unit and space occupied by the unit are neglected because of lack of required data during this conceptual design phase. The formulation of FEDI and TDI is stated in Equations (7)–(16)

$$F_1 = 0.1M \times \frac{Hc}{K} \tag{7}$$

$$F_2 = 1.304 \times 10^{-3} \times Pp \times V \tag{8}$$

$$F_3 = 1 \times 10^{-3} \times \frac{1}{(T+273)} \times (Pp - Vp)^2 \times V \tag{9}$$

$$F_4 = M \times \frac{Hr \times n}{K} \tag{10}$$

$$Dp = (F_1 \times pn_1 + F \times pn_2 + F_4 \times pn_7) \times pn_3 \times pn_4 \tag{11}$$

$$FEDI = 4.76(Dp)^{1/3} \tag{12}$$

$$G = A \times m \tag{13}$$

$$TDI = a(G \times pn_1 \times pn_2 \times pn_3 \times pn_4)^b \tag{14}$$

$$FEDI_{total} = \sum_j FEDI_j + \sum_k FEDI_k \tag{15}$$

$$TDI_{total} = \sum_j TDI_j + \sum_k TDI_k \tag{16}$$

Here, $F_1$, $F_2$, $F_3$, and $F_4$ are chemical, physical, and reaction energy, respectively. $M$ is the mass flow rate of the chemical (kg/s), $Hc$ is the heat of combustion of the chemical (kJ/kg), and $K$ is a constant (3.148). In Equations (8) and (9), $Pp$ (kPa) and $Vp$ (kPa) are the process pressure and vapor pressure of the chemical at process temperature. $V$ is the volumetric flow rate of chemical (m³/h). *TDI* involves a $G$ factor and several penalties. $G$ factor is obtained from $A$ (phase condition), and $m$ is the anticipated release rate in kg/s.

*2.4. Constraints*

Biomass availability constraint is shown in Equation (17).

$$EFB_{\min} \leq EFB \leq EFB_{\max} \tag{17}$$

This shows that EFB consumed in the production process is within a minimum and maximum range of EFB supplied. EFB is delivered to plant using a truck with a capacity of 25 ton/truck [64]. The maximum allowable number of trucks to transport EFB is taken to vary from 2 trucks/h to 4 trucks/h. This range is set based on the assumed biomass unloading time of 15–30 min/truck and only one unloading facility within the plant. The assumption of 15 min/truck is based on about 5–10 min of unloading time, and the remaining time is for traveling in and out of the plant. This number becomes the logistic limit within the plant, which in turn limits the maximum production to 4 trucks/h. The same reasoning applies to 2 trucks/h.

Chemical mass yield information is gathered from the experimental data of references and they are modelled in min and max scenarios, which are used in the process simulation, as stated in Equation (18).

$$X_{k\min} \leq X_k \leq X_{k\max} \tag{18}$$

The mass yield of each process $X_k$ is set within the range of $X_{k\min}$ and $X_{k\max}$, which are determined from the experimental data of xylitol [49,50], levulinic acid [51], succinic acid [53], guaiacol [53], and vanillin [54]. The yield information is modelled in min and max scenarios which are used in the process simulation. The data are tabulated in Table 1.

**Table 1.** The mass yield of each process.

| Product | Min Yield (wt%) | Max Yield (wt%) |
|---|---|---|
| Xylitol | 96 | 99 |
| Levulinic acid | 67 | 77 |
| Succinic acid | 73 | 80 |
| Guaicol | 33 | 37 |
| Vanillin | 7 | 7.7 |

The biochemical products are expected to fulfill the global demand for each product, as shown in Equation (19) and tabulated in Table 2.

$$P_m \geq Demand \tag{19}$$

The products obtained from the integrated biorefinery $P_m$ must be equal to or greater than the global demand.

**Table 2.** The global demand for each product.

| Chemical | Global Demand (t/yr) | Ref |
|---|---|---|
| Xylitol | 266,500 | [65] |
| Levulinic acid | 14,950 | [66] |
| Succinic acid | 710,000 | [67] |
| Guaiacol | 45,000 | [68] |
| Vanillin | 37,286 | [69] |

*2.5. Genetic Algorithm*

Genetic algorithm (GA) is a directed random search technique that is modeled on the natural evolution/selection process toward the survival of the fittest [70]. In briefly, this algorithm begins with an initial population of chromosomes or trial solutions or individuals, each characterized by a set of values of decision variables in MOO problem. The individuals are generated randomly within the lower and upper bounds of decision variables, and are supplied to calculate objective functions (annual profit, GWP, FEDI, and TDI). Individuals from one generation are used to create a new population, based on selection and reproduction by crossover and mutation. The crossover is carried out on randomly selected parent individuals and generates offspring by swapping parts of the parent chromosomes. Afterward, mutation occurs by randomly altering the offspring created by crossover. This drastic change helps to prevent solutions from being stuck near a local optimum. Thus, the algorithm generates different individuals for the next generation (i.e., new population) from the previous population. These steps are repeated until some termination criterion (e.g., maximum number of generations or improvement of the best solution) is satisfied.

One adaptation of GA for MOO is gamultobj code in MATALAB, which was used in this study. Table 3 lists the values of GA parameters used in the present study. Bounds on the decision variables are tabulated in Table 4. The developed model equations used in multi-objective genetic algorithm are given in Section B in the Supplementary Materials.

**Table 3.** Parameter setting of gamultiobj algorithm.

| Specified Option | Value |
| --- | --- |
| Solver | gamultiobj |
| Number of generations, *Ngen* | 1000 |
| Population size, *Npop* | 100 |
| Crossover probability, *Pc* | 0.8 |
| Mutation probability, *Pm* | 0.2 |
| Number of variables | 8 |

**Table 4.** The decision variables of the integrated biorefinery.

| Decision Variables | Lower Bound | Upper Bound |
| --- | --- | --- |
| EFB (ton/h) | 50 | 100 |
| Xylitol yield (wt%) | 96 | 99 |
| Mass allocation of glucose | 0 | 1 |
| Levulinic acid yield (wt%) | 67 | 77 |
| Succinic acid yield (wt%) | 73 | 80 |
| Mass allocation of lignin | 0 | 1 |
| Guaiacol yield (wt%) | 33 | 37 |
| Vanillin yield (wt%) | 7 | 7.7 |

*2.6. Multi-Criteria Decision Making*

To choose one preferred solution from the Pareto-optimal frontier, FAHP and TOPSIS are applied. Both these are outlined in this section.

2.6.1. Fuzzy Analytical Hierarchy Process

FAHP is a combination of fuzzy logic and linguistic variables [71]. In this work, such a method is applied to overcome the uncertainty in the judgment of decision makers in their selection of the best optimum solution [72]. The steps involved in the computation of criterion weights using FAHP are described in this sub-section.

1.     Generating a hierarchy system evaluation

This work is to find the best optimal biorefinery superstructure configuration incorporating the sustainability criteria, which are comprised of economic, environment and safety aspects. The Pareto-optimal solutions generated by solving the MOO problem are the alternatives. The detailed decision model is presented in Figure 4.

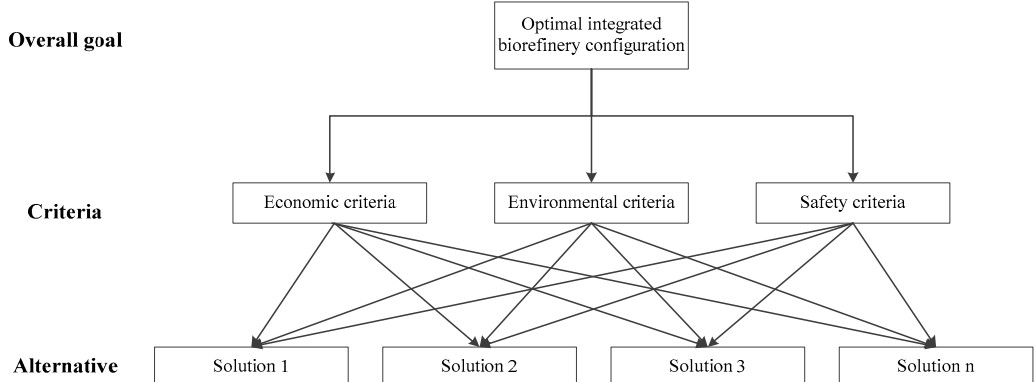

**Figure 4.** Schematic of alternate Pareto-optimal solutions for the integrated biorefinery.

2.     Create a fuzzified pairwise comparison matrix

In this work, the decision maker's judgment is gathered according to the questionnaire, which was taken from Yan [73] and is summarized in Table S6 in the Supplementary Materials. Afterward, the pairwise comparison of the evaluation criteria is made based on a linguistic scale. The calibrated linguistic fuzzy scale suggested by Promentilla et al. [74] is implemented in this work, as shown in Table 5. The result of fuzzy pairwise comparison judgments can be expressed in the form of a matrix. The matrix A is an m x m real matrix, where *m* is the number of evaluation criteria. Each entry $a_{jk}$ of a matrix A represents the importance of the criterion. If $a_{jk} > 1$, the *j*th criterion is more important than the *k*th criterion; if $a_{jk} < 1$, the *j*th criterion is less important than the *k*th criterion. Then, the weight of the criteria and the rating of alternatives concerning can be calculated using Equation (20).

$$\widehat{A_k} = \begin{pmatrix} (1,1,1) & \widehat{a_{12k}} & \cdots & \widehat{a_{1nk}} \\ \widehat{a_{21k}} & (1,1,1) & \cdots & \widehat{a_{2nk}} \\ \vdots & \vdots & \ddots & \vdots \\ \widehat{a_{m1k}} & \widehat{a_{m2k}} & \cdots & (1,1,1) \end{pmatrix} \text{ where } \widehat{a_{ijk}} = \frac{1}{\widehat{a_{ijk}}} = (\frac{1}{a_{ijk}^U}, \frac{1}{a_{ijk}^M}, \frac{1}{a_{ijk}^L}) \quad (20)$$

**Table 5.** The global demand for each product.

| Fuzzy Number | Linguistic Term | Symbol | Triangular Fuzzy Numbers (TFNs) |
|:---:|:---:|:---:|:---:|
| 1 | Equally | EQ | (1.0, 1, 1.0) |
| 2 | Slightly More | SM | (1.2, 2, 3.2) |
| 3 | Moderately More | MM | (1.5, 3, 5.6) |
| 5 | Strongly More | ST | (3.0, 5, 7.9) |
| 8 | Very Strongly More | VS | (6.0, 8, 9.5) |

3.     Aggregation of DM's judgment

Then, fuzzy judgments of all DMs are aggregated using Equation (21):

$$a_{ij}^L = (\prod_{K=1}^{K} a_{ijk}^L)^{vk}; a_{ij}^M = (\prod_{K=1}^{K} a_{ijk}^M)^{vk}; a_{ij}^U = (\prod_{K=1}^{K} a_{ijk}^U)^{vk} \quad (21)$$

4.　Computation of criteria weights

The criteria weights are then deduced through the non-linear programming model proposed by Tan et al. [75], as stated in Equations (22) and (23). This model approximates the criteria weights by constraining the consistency index ($\lambda$) within the fuzzy bound. A positive $\lambda$ indicates a consistent fuzzy pairwise judgment given by the expert; hence, the criteria weights elicited are acceptable and can be applied for further computation. Note that $\lambda = 1$ suggests perfect consistency in preserving the order of preference intensities. Maximize $\lambda$ Subject to

$$\lambda(a_{ij}^M - a_{ij}^L)w_j - w_i + a_{ij}^L w_j \leq 0, i = 1, \ldots, m-1, j = 2, \ldots, n \tag{22}$$

$$\lambda(a_{ij}^U - a_{ij}^M)w_j + w_i - a_{ij}^U w_j \leq 0, i = 1, \ldots, m-1, j = 2, \ldots, n \tag{23}$$

$\sum_{j=1}^{n} w_j = 1$, where $w_j > 0$

2.6.2. Technique for Order Preference by Similarity to Ideal Solution

The methodology of TOPSIS is briefly explained below. Application and assessment of TOPSIS to many problems are available in Wang et al. [76].

1.　Calculate a normalized matrix

The decision matrix of $X_{ij}$ is normalized using the following Equation (24):

$$X_{ij} = \frac{X_{ij}}{\sqrt{\sum_{j=1}^{n} X_{ij}^2}}, j = 1, 2, 3, \ldots, j \; i = 1, 2, 3, \ldots, i \tag{24}$$

2.　Calculate weighted normalized matrix

The normalized decision matrix is multiplied by the weight, $w_i$, obtained from Equations (22) and (23). Calculation is performed by the following Equation (25):

$$v_{ij} = w_i \times X_{ij}, j = 1, 2, 3, \ldots, j \; i = 1, 2, 3, \ldots, i \tag{25}$$

3.　Calculate the ideal best and ideal worst value

The positive ideal solution $A^*$ and the negative ideal solution $A^-$ are determined using the following Equations (26) and (27):

$$A^* = \{v_1^*, \ldots, v_i^*\} = \{(\max v_{ij} | i \in I'), (\min v_{ij} | i \in I'')\} \tag{26}$$

$$A^- = \{v_1^-, \ldots, v_i^-\} = \{(\min v_{ij} | i \in I'), (\max v_{ij} | i \in I'')\} \tag{27}$$

4.　Calculate Euclidean distance from the positive/negative ideal

The distance from the positive and negative ideal for each alternative can be computed by the Euclidean distance as given in Equations (28) and (29).

$$D_j^* = \sqrt{\sum_{i=1}^{n} \left(v_{ij} - v_{ij}^*\right)^2}, j = 1, 2, 3, \ldots, j \tag{28}$$

$$D_j^- = \sqrt{\sum_{i=1}^{n} \left(v_{ij} - v_{ij}^-\right)^2}, j = 1, 2, 3, \ldots, j \tag{29}$$

5. Calculate performance score

The relative closeness of the *i*th alternative is defined as

$$CC_j^* = \frac{D_j^-}{D_j^* + D_j^-}, j = 1, 2, 3, \ldots, j \tag{30}$$

The ranking priority of alternatives is determined based on the higher score of $CC_j^*$.

## 3. Results and Discussion

Prior to MOO, a sustainability assessment of each process was conducted to see its economy, safety, and environment impact. These results are summarized in Tables S4 and S5 and Figure S1 in the Supplementary Materials. Three cases of MOO are explored in this work. They are two cases of bi-objective (maximize annual profit and minimize GWP, and maximize annual profit and minimize hazard potentials) and one case of tri-objective (maximize the annual profit and minimize GWP, FEDI, and TDI), as shown in Table 6. The considered constraints are applied in all cases. Selection of one optimal solution from the non-dominated solutions is employed in case of tri-objective optimization as it includes all three pillars of sustainability. This step of MOO was adapted from Rangaiah et al. [77], and the MS Excel-based program developed by Wang and Rangaiah [76] was employed as a tool for multi-criteria decision-making. The MOO problem was formulated in MATLAB and solved using the multi-objective genetic algorithm code (gamultiobj).

**Table 6.** Multi-objective optimization problem formation for optimizing integrated biorefinery configuration.

| Case Study | Objective Functions |
|---|---|
| MOO Case 1 | Maximize profit and Minimize Environmental impact |
| MOO Case 2 | Maximize profit and Minimize Hazard potential |
| MOO Case 3 | Maximize profit, Minimize Environmental impact and Minimize Hazard potential |

### 3.1. Case 1: Maximize Profit and Minimize Global Warming Potential

Figure 5 shows the Pareto-optimal solutions for Case 1 bi-objective optimization, where the x-axis represents the annual profit and y-axis is GWP. Each Pareto-optimal point represents the optimal design of the biorefinery superstructure with a unique combination (trade-off) between the annual profit and GWP. Three solutions (marked S1, S2, and S3 in Figure 5) on the Pareto-optimal front are chosen as examples of optimal solutions for further analysis. Table 7 shows the decision variables at the maximum (S1), intermediate (S2), and minimum (S3) points of the Pareto-optimal front, and Table 8 presents the main results at these optimal solutions. It can be seen that the environmental impact is decreasing from S1 to S2 to S3 at the expense of decreasing annual profit. Ideally, the profit has to be maximized and the GWP minimized simultaneously. However, as depicted in the Pareto-optimal front, the improved performance of one objective function is obtained only at the expense of deteriorating another objective function. In this manner, Figure 5 has 35 optimal solutions, and each of them is equally good.

Point S1 denotes the solution with the highest annual profit and GWP. In this solution, the supply of EFB and mass yield of process reaches the maximum values, as well as the mass allocation of cellulose, hemicellulose, and lignin. In particular, cellulose is allocated to producing levulinic acid (10%) and succinic acid (90%), while lignin is to produce guaiacol (70%) and vanillin (30%). Here, the allocation of cellulose to succinic acid is higher due to greater global demand for it, even though levulinic acid production shows better performance in economy and environmental impact due to having a lower capital investment and annual production cost and higher product yield and selling price as well as lower GWP score. The same goes to producing more guaiacol than vanillin because of higher global demand. There is 30% of lignin used for vanillin production due to having better

performance in economy and environmental impact. Xylose (obtained by conversion of xylan contained in hemicellulose by dilute acid pretreatment) is allocated 100% to produce xylitol since it has no other option. Figure 6 shows the integrated biorefinery configuration of solution S1 where 100 ton/h of EFB is supplied to the process, which corresponds to the maximum EFB supplied to the plant. This amount of EFB produces 18.9 ton/h xylitol, 1.77 ton/h levulinic acid, 22.1 ton/h succinic acid, 5.04 ton/h guaiacol, and 0.5 ton/h vanillin. Total resulting annual profit and GWP scores are 923 M USD/year and 285 tonCO$_2$-eq.

Point S2 is an intermediate solution, where a slight decrease of capacity (99 ton/h) results in lower profit by 1.95% and also GWP by 2%. The corresponding biorefinery configuration has annual profit and GWP of 905 M USD/year and 283 tonCO$_2$-eq, respectively. Like in point S1, 10% cellulose is sent to produce levulinic acid while the remaining goes to produce succinic acid, and 70% of lignin is to produce guaiacol and the rest goes to vanillin. Finally, point S3 represents the lowest of both profit and GWP. Here, the allocation of cellulose and lignin are similar to points S1 and S3. The S3 solution gives an annual profit of 880 M USD/year and GWP of 281 tonCO$_2$-eq, where the required EFB is supplied at 97 ton/h to produce xylitol (18.4 ton/h), levulinic acid (1.37 ton/h), succinic acid (22 ton/h), guaiacol (5.01 ton/h), and vanillin (0.45 ton/h), as depicted in Figure 7. It is clear that the allocation of cellulose and lignin show similar percentage in all Pareto-optimal solutions. With slightly higher capacity, both profit and higher environmental impact are higher. The plant capacity, on the other hand, has EFB supply truck as its bottleneck. Hence, to increase profit, more EFB supply trucks have to be organized, which leads to a logistic issue within the plant.

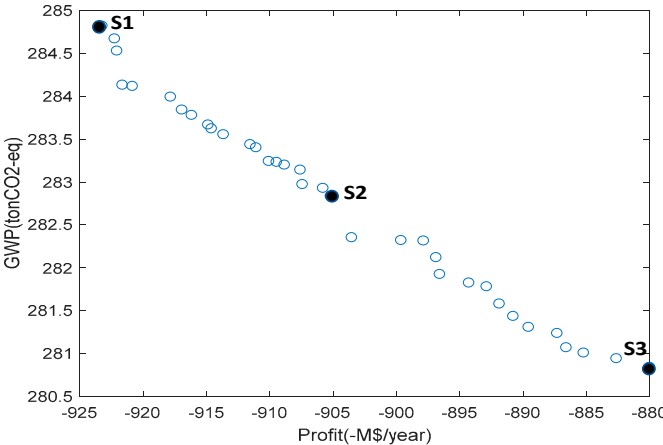

**Figure 5.** Pareto-optimal front for profit versus global warming potential.

**Table 7.** Decision variables that correspond to the selected Pareto-optimal points in Figure 5.

| Decision Variables | S1 | S2 | S3 |
|---|---|---|---|
| EFB (ton/h) | 100 | 99 | 97 |
| Xylitol yield (wt%) | 99 | 99 | 99 |
| Mass allocation of glucose | 0.1 | 0.1 | 0.1 |
| Levulinic acid yield (wt%) | 73 | 73 | 73 |
| Succinic acid yield (wt%) | 80 | 80 | 80 |
| Mass allocation of lignin | 0.7 | 0.7 | 0.7 |
| Guaiacol yield (wt%) | 36 | 36 | 35 |
| Vanillin yield (wt%) | 7.6 | 7.6 | 7.6 |

**Table 8.** Optimal objective values corresponding to the selected Pareto-optimal points in Figure 5.

| | S1 | S2 | S3 |
|---|---|---|---|
| Annual profit (M$/year) | 923 | 905 | 880 |
| GWP (tonCO$_2$-eq) | 285 | 283 | 281 |

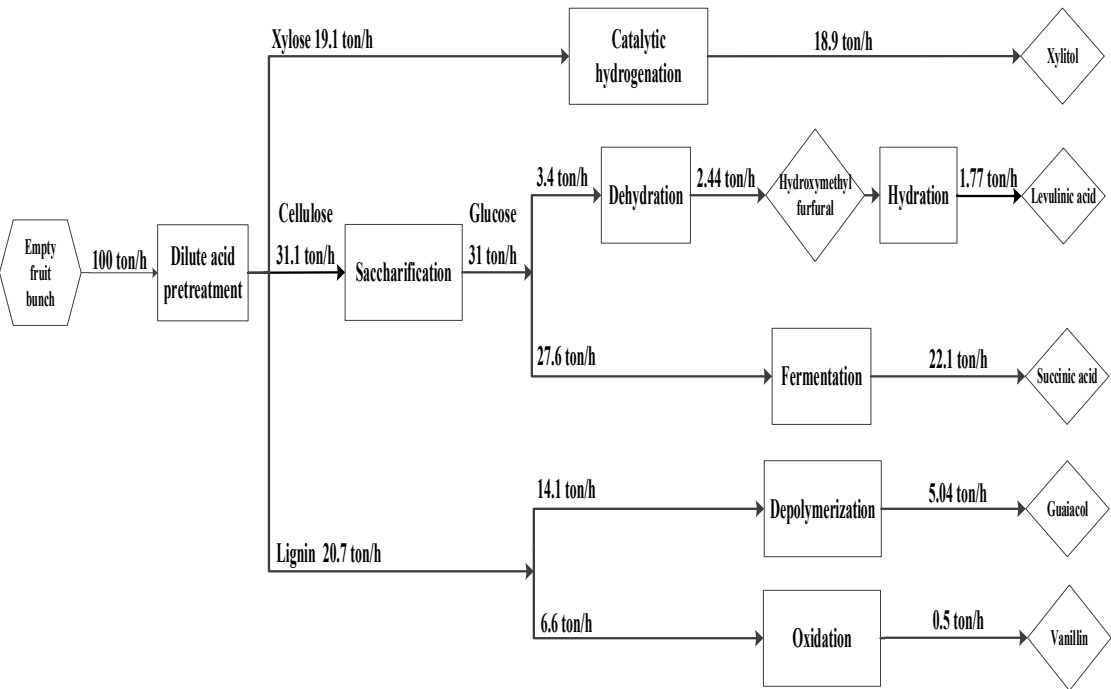

**Figure 6.** The biorefinery configuration of the optimal point S1 for profit and global warming potential objectives.

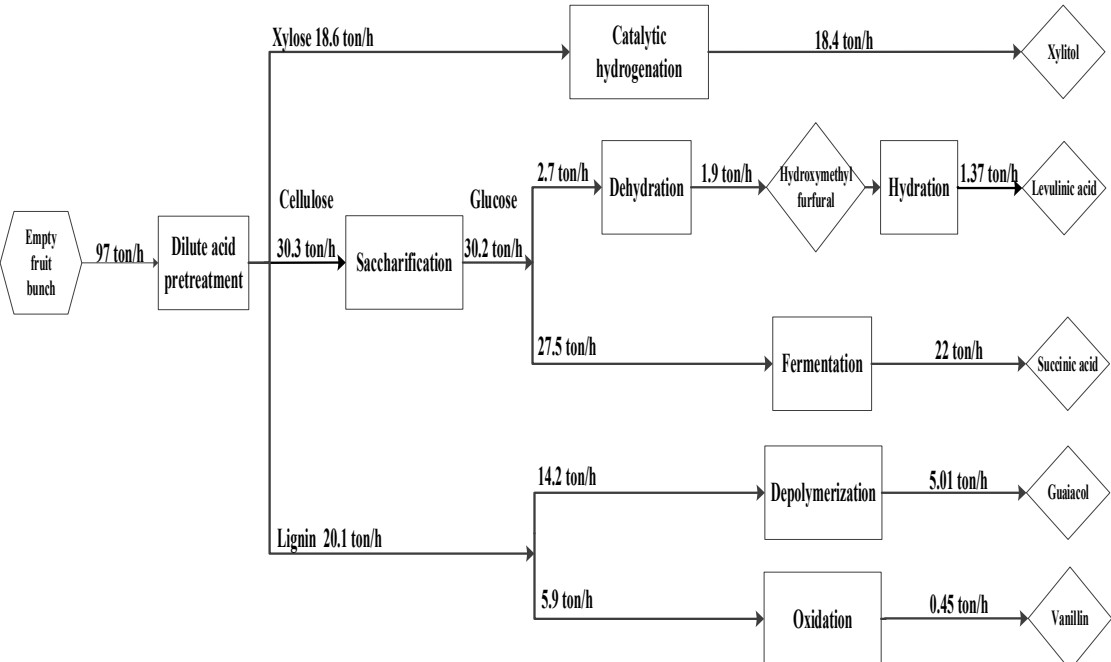

**Figure 7.** The biorefinery configuration of the optimal point S3 for profit and global warming potential objectives.

### 3.2. Case 2: Maximize Profit and Minimize Hazard Potential

This scenario considers the maximization of profit and minimization of FEDI and TDI. Figure 8a,b represents the Pareto-optimal curve for annual profit versus FEDI and annual profit versus TDI from the MOO results. Three solutions on each curve are identified for comparison. Tables 9 and 10 give the corresponding sets of optimal decision variables and objectives, respectively. For profit and FEDI scenario, point S1 represents the solution with the highest of both annual profit and FEDI score of

923 M USD/year and 592, respectively. In this solution, the maximum of 100 ton/h of EFB is necessary with the same allocation of cellulose as in Case 1 (all three solutions in Figure 5). Interestingly, lignin allocation is now different, where 80% is sent to guaiacol and the remaining goes to vanillin. This is because guaiacol production has lower FEDI score than vanillin. The use of ethyl acetate in extraction step increases FEDI score of vanillin process. Figure 9 shows biorefinery configuration for solution S1 in Figure 8a. This biorefinery fulfills global demand of xylitol by 57%, levulinic acid by 95%, succinic acid by 25%, guaiacol by 100% and vanillin by only 9%.

Point S2 as a middle solution in Figure 8 yields annual profit and FEDI of 864 M USD/year and 577, respectively. The same allocation of cellulose is obtained here, while all lignin is allocated to produce guaiacol due to lower FEDI score than that of vanillin process. In point S3, the lowest annual profit and FEDI score are obtained, which are 789 M USD/year and 570, respectively. The cellulose allocation is only 3% to levulinic acid, and the rest goes to succinic acid because levulinic acid production has a higher flammability and explosion risk compared to the production of succinic acid. This is due to severe operating conditions and hazardous chemicals involved in levulinic acid production. On the other hand, lignin is fully allocated to guaiacol production as it results a lower FEDI score than vanillin production. The lower solution (S3 in Figure 8a) on the Pareto-optimal curve has the lower FEDI score of guaiacol. Its configuration in Figure 10 shows that 96 ton/h of EFB is supplied to the biorefinery to produce xylitol (18 ton/h), levulinic acid (0.43 ton/h), succinic acid (22.5 ton/h), and guaiacol (6.6 ton/h).

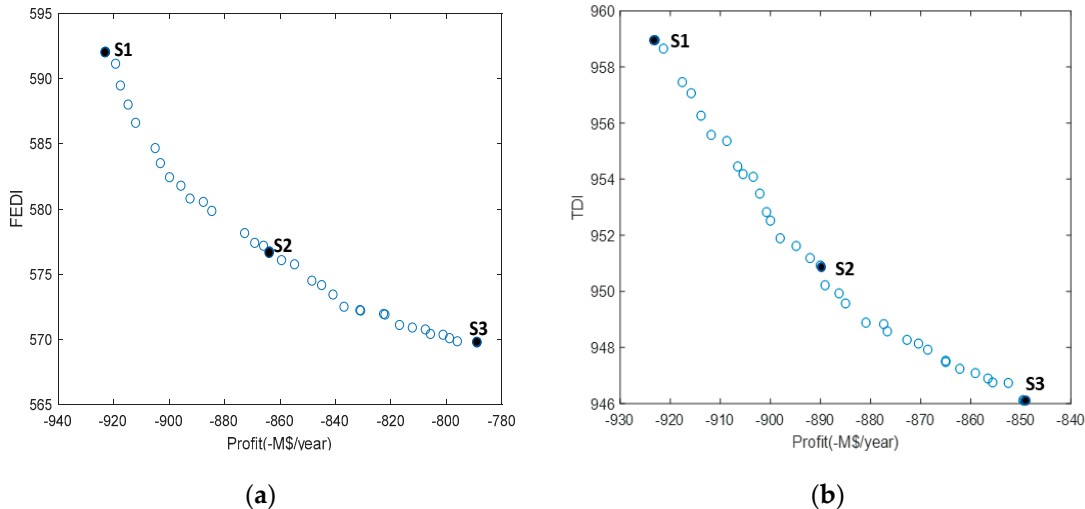

(**a**)　　　　　　　　　　　　　　(**b**)

**Figure 8.** Pareto-optimal fronts for (**a**) profit versus fire explosion damage index and (**b**) profit versus toxicity damage index.

**Table 9.** Decision variables that correspond to the selected optimal solutions in Figure 8.

| Decision Variables | Profit and FEDI | | | Profit and TDI | | |
|---|---|---|---|---|---|---|
| | **S1** | **S2** | **S3** | **S1** | **S2** | **S3** |
| EFB (ton/h) | 100 | 98 | 96 | 100 | 98 | 97 |
| Xylitol yield (wt%) | 99 | 98 | 98 | 99 | 99 | 97 |
| Mass allocation of glucose | 0.1 | 0.1 | 0.03 | 0.1 | 0.1 | 0.1 |
| Levulinic acid yield (wt%) | 75 | 74 | 73 | 74 | 74 | 71 |
| Succinic acid yield (wt%) | 80 | 79 | 77 | 80 | 80 | 79 |
| Mass allocation of lignin | 0.8 | 1 | 1 | 0.7 | 0.7 | 0.7 |
| Guaiacol yield (wt%) | 36 | 33 | 33 | 36 | 36 | 36 |
| Vanillin yield (wt%) | 7.7 | 7.5 | 7.6 | 7.7 | 7.6 | 7 |

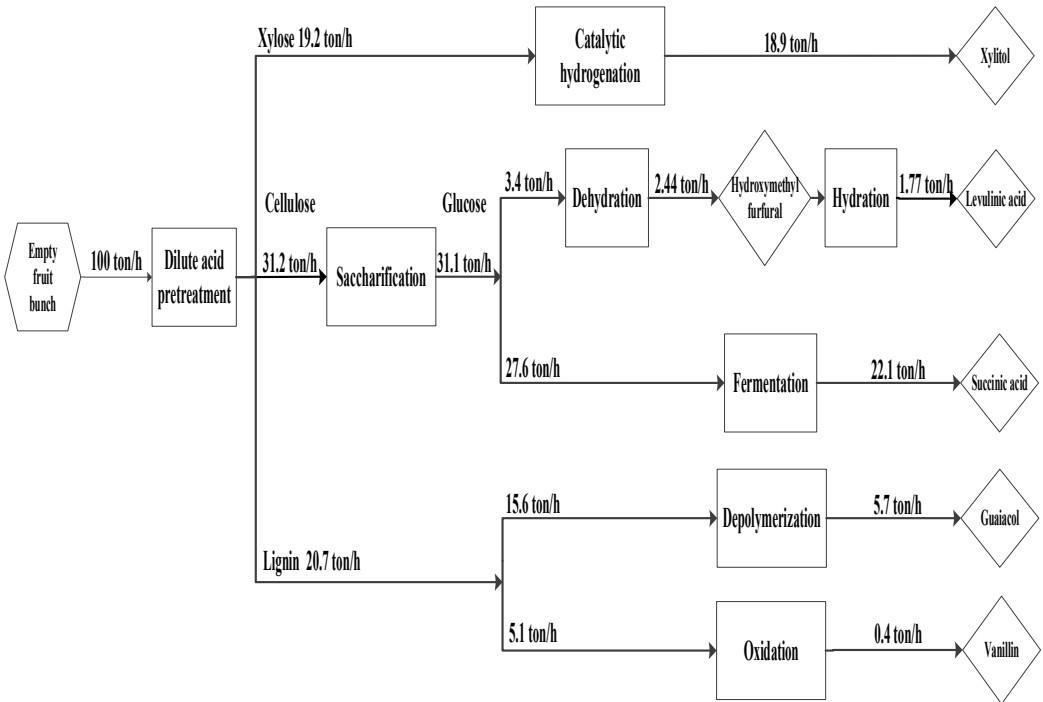

**Figure 9.** The optimal biorefinery configuration of the optimal point S1 for profit and fire explosion damage index in Figure 8a.

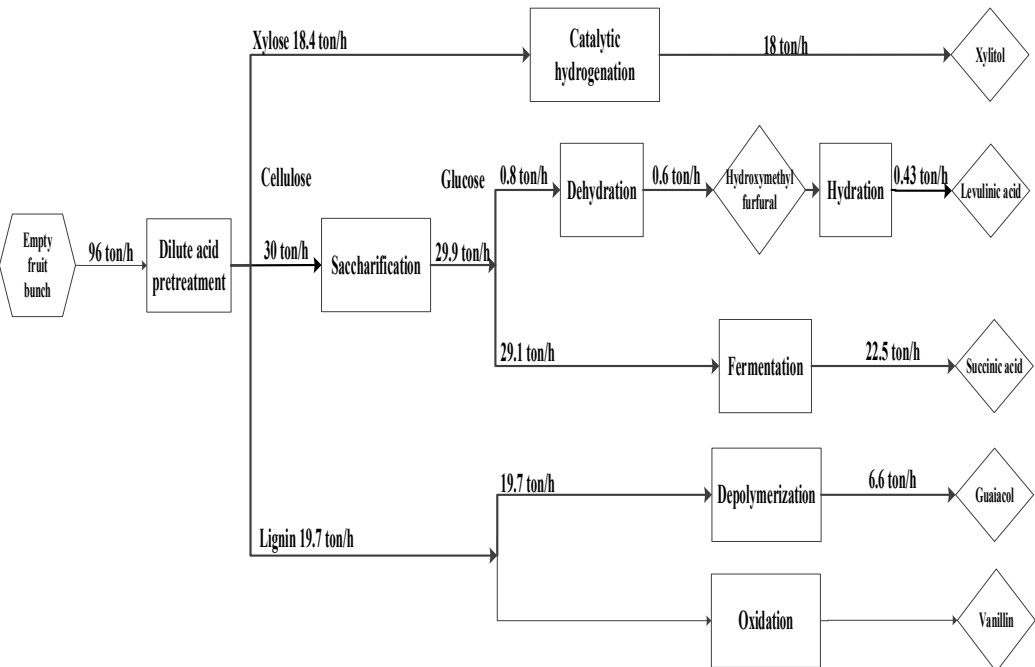

**Figure 10.** The optimal biorefinery configuration of the optimal point S3 for profit and fire explosion damage index in Figure 8a.

**Table 10.** Optimal objective values corresponding to the selected Pareto-optimal points in Figure 8.

|  | Profit and FEDI | | | Profit and TDI | | |
|---|---|---|---|---|---|---|
|  | **S1** | **S2** | **S3** | **S1** | **S2** | **S3** |
| Annual profit (M$/year) | 923 | 864 | 789 | 923 | 889 | 850 |
| FEDI | 592 | 577 | 570 | - | - | - |
| TDI | - | - | - | 959 | 950 | 946 |

For-profit and TDI scenario (Figure 8b), point S1 presents the solution with the highest annual profit of 923 M USD/year and TDI score of 959. In this solution, maximum of 100 ton/h of EFB is required and mass allocation of sugars to levulinic acid and succinic acid are 10% and 90%, respectively. In this regard, succinic acid has major allocation because of higher global demand. TDI score of succinic acid process is higher than that of levulinic acid process due to high operating conditions in the purification section and succinic acid is categorized as a toxic/corrosive chemical. However, sustainability assessment of levulinic acid in terms of annual profit and TDI shows better result than succinic acid. Thus, a small percentage (10%) of cellulose goes to levulinic acid. Furthermore, 70% of lignin is allocated to guaiacol and the rest to vanillin. Like succinic acid, guaiacol has higher global demand but lower annual profit and TDI score. Hence, vanillin has gained 30% of lignin allocation due to better economics and safety in terms of TDI. Figure 11 shows the selected biorefinery configuration (S1 in Figure 8b) to produce 18.8 ton/h of xylitol, 1.78 ton/h of levulinic acid, 22 ton/h of succinic acid, 5.16 ton/h of guaiacol, and 0.49 ton/h of vanillin. It can fulfill the global demand of xylitol by 56%, levulinic acid by 95%, succinic acid by 25%, guaiacol by 92%, and vanillin by 11%.

Point S2 in Figure 8b is an intermediate solution, where the total profit and TDI are 889 M USD/year and 950, respectively. The same trend of materials allocation (as for point S1) is seen here. Point S3 in Figure 8b gives the lowest of both profit and TDI of 850 M USD/year and 946, respectively; it also has the same mass allocation of materials. Figure 12 shows the configuration for point S3, where 97 ton/h of EFB is supplied to the biorefinery producing xylitol (18 ton/h), levulinic acid (1.11 ton/h), succinic acid (22 ton/h), guaiacol (5 ton/h), and vanillin (0.45 ton/h). It fulfills the global demand of xylitol by 54%, levulinic acid by 59%, succinic acid by 25%, guaiacol by 89%, and vanillin by 10%. It is interesting to note that all solutions in this case are similar to those in Case 1 since levulinic acid and vanillin have better performance in both economy and TDI.

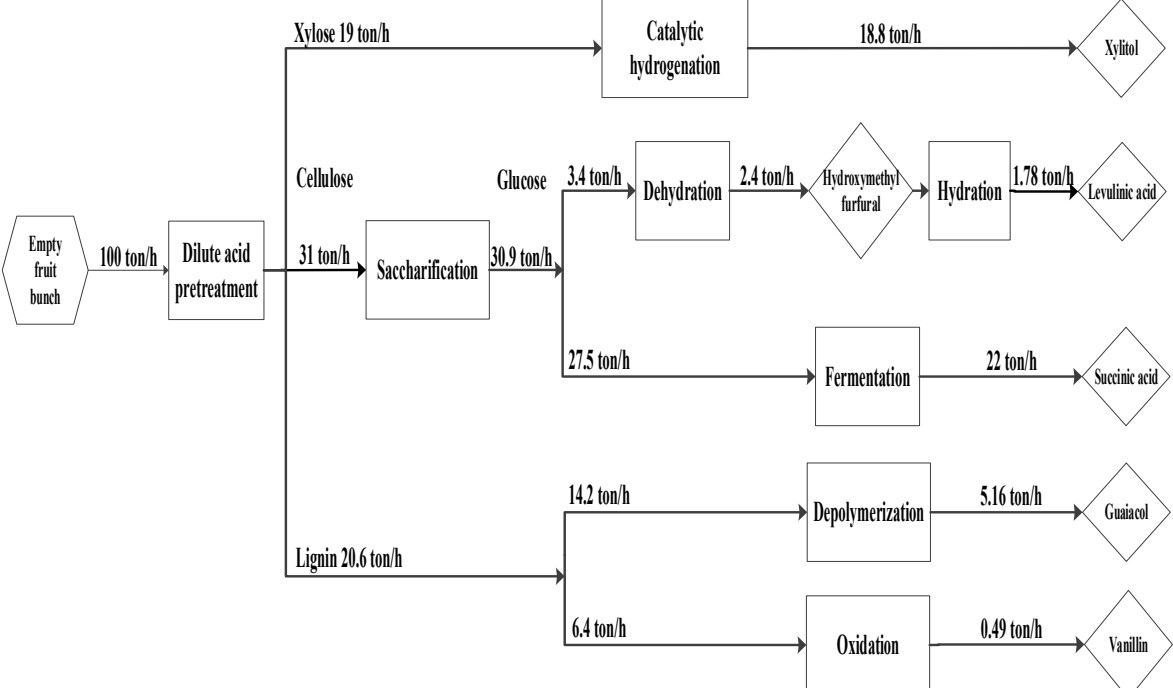

**Figure 11.** The optimal biorefinery configuration of the optimal point S1 for profit and toxicity damage index in Figure 8b.

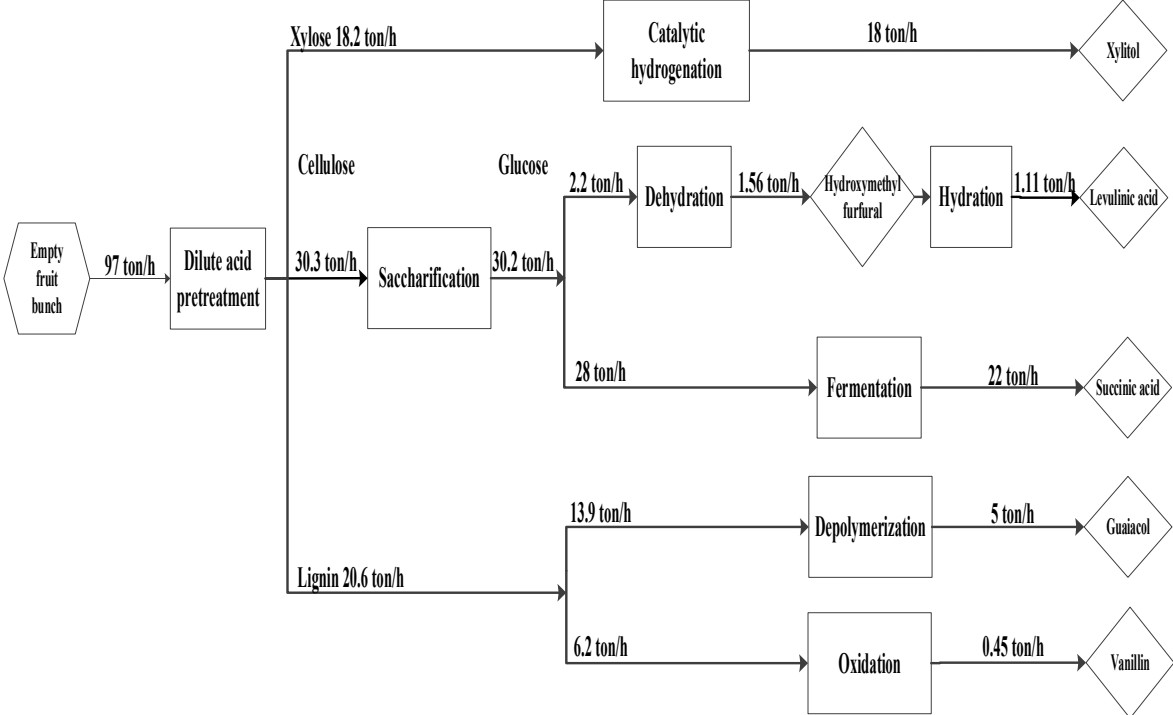

**Figure 12.** The optimal biorefinery configuration of the optimal point S3 for profit and toxicity damage index in Figure 8b.

### 3.3. Case 3: Maximize Profit, Minimize Environmental Impact, and Minimize Hazard Potential

From the Pareto-optimal solutions for Case 3 (Figure 13), one preferred solution is selected through the combination of FAHP and TOPSIS. The subjective weight of different evaluation criteria for this study was from Yan [75], where five experts were asked to give the weight of each criterion. Each weight was compiled to make the fuzzy pairwise comparison. Then, fuzzy geometric mean was used to calculate the weights/preferences of decision-makers for each criterion as tabulated in Table 11. The weights obtained for each criterion were finally 0.5 for the economy (C1), 0.2 for the environment (C2), and 0.3 for safety (C3), as listed in Table 12. It can be seen that the economic criterion would be the main factor in the decision making compared to the other two. It is interesting to note that safety has relatively more weightage than environment. The weights (Table 12) determined using the fuzzy-AHP method were then used for ranking the alternative solutions by TOPSIS. Calculations of TOPSIS are summarized in Tables S7 and S8 in the Supplementary Materials.

**Table 11.** Fuzzy geometric mean of weights/preferences.

| Criteria | C1 | C2 | C3 |
|----------|-----|-----|-----|
| C1 | (1,1,1) | (1.23,1.66,2.2) | (1.04, 1.72, 2.78) |
| C2 | (0.45, 0.6, 0.81) | (1,1,1) | (0.39,0.64,1.11) |
| C3 | (0.36, 0.58, 0.96) | (0.90, 1.55, 2.59) | (1,1,1) |

**Table 12.** Weight matrix.

| Criteria | Weight |
|----------|--------|
| C1 | 0.5 |
| C2 | 0.2 |
| C3 | 0.3 |

Figure 13a shows the Pareto-optimal solutions for annual profit, GWP, and GEDI, whereas Figure 13b shows them for annual profit, GWP, and TDI. The optimal values for the chosen solution (shown by filled circle in both these plots) by FAHP and TOPSIS are summarized in Tables 13 and 14. Since the main factor is economy, the preferred solution out of many Pareto-optimal solutions is the one with the highest profit. The corresponding optimal biorefinery configuration is depicted in Figure 14.

For the chosen optimal configuration, annual profit, GWP, FEDI, and TDI are 932 M$/year, 284 tonCO$_2$-eq, 595, and 957, respectively. This configuration is at the maximum capacity of 100 ton/h of EFB and the mass allocation of glucose is 10% to levulinic acid and the remaining to succinic acid, whereas 70% of lignin goes to guaiacol and the rest to vanillin. The xylitol production is 18.9 ton/h, which fulfills the global demand by 55%. The production of 1.83 ton/h of levulinic acid fulfills 98% of global demand, 22 ton/h of succinic acid fulfills 25% of demand, 5 ton/h of guaiacol fulfills ~90% of demand and 0.55 ton/h of vanillin fulfills ~12% of demand. As expected, maximum profit corresponds to maximum EFB that can be supplied to the plant.

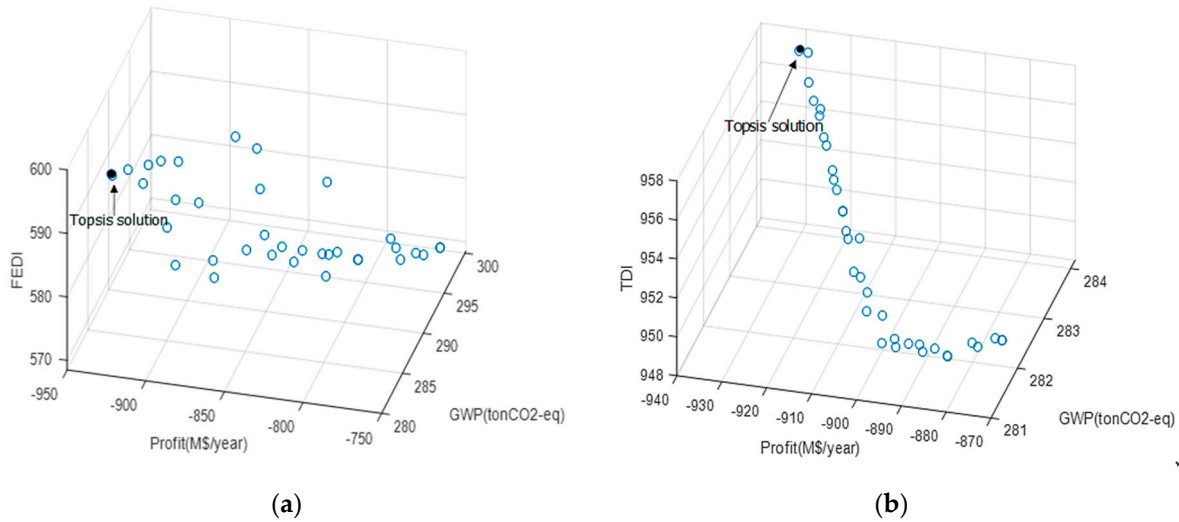

(**a**)　　　　　　　　　　　　　　　　　　　(**b**)

**Figure 13.** Pareto-optimal fronts for (**a**) profit, global warming potential, and fire explosion damage index and (**b**) profit, global warming potential, and toxicity damage index.

**Table 13.** Decision variables that correspond to the selected Pareto-optimal solutions in Figure 13.

| Decision Variables | Profit, GWP and FEDI | Profit, GWP and TDI |
|---|---|---|
| EFB (ton/h) | 100 | 100 |
| Xylitol yield (wt%) | 99 | 99 |
| Mass allocation of glucose | 0.1 | 0.1 |
| Levulinic acid yield (wt%) | 73 | 73 |
| Succinic acid yield (wt%) | 80 | 80 |
| Mass allocation of lignin | 0.7 | 0.7 |
| Guaiacol yield (wt%) | 37 | 37 |
| Vanillin yield (wt%) | 7.7 | 7.7 |

**Table 14.** Optimal objective values for the selected Pareto-optimal solutions in Figure 13.

| Objective | Profit, GWP and FEDI | Profit, GWP and TDI |
|---|---|---|
| Profit (M$/year) | 932 | 931 |
| GWP (tonCO$_2$-eq) | 284 | 284 |
| FEDI | 595 | - |
| TDI | - | 957 |

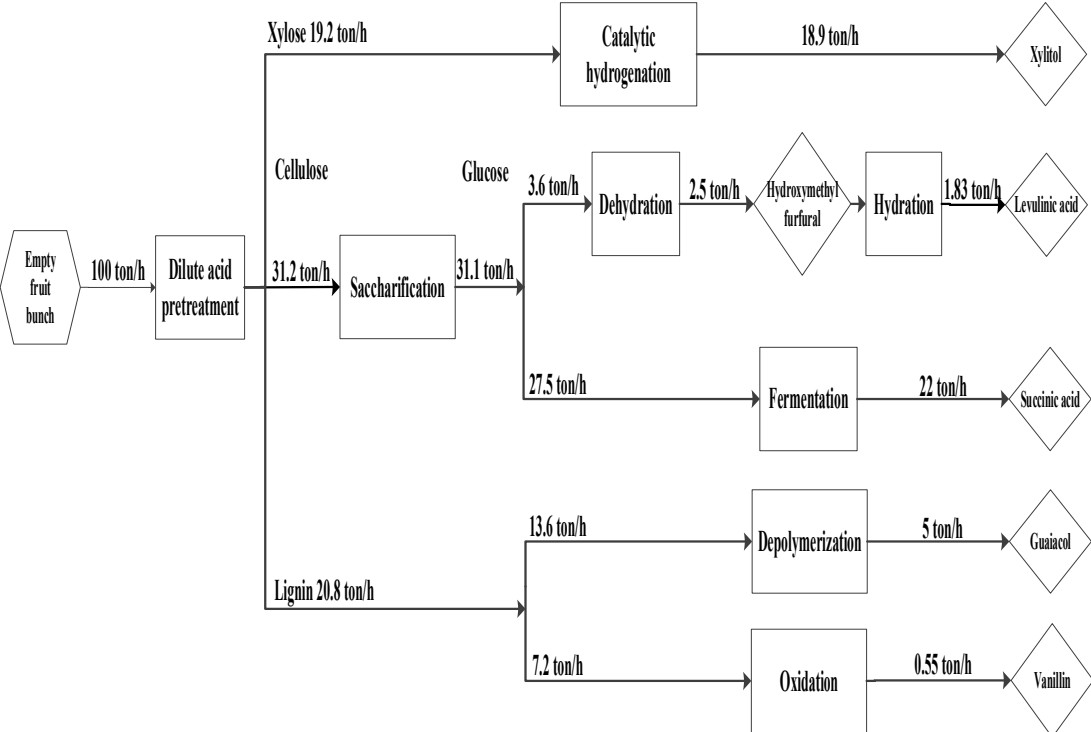

**Figure 14.** The optimal biorefinery configuration for simultaneously maximizing annual profit and minimizing global warming potential, fire explosion damage index, and toxicity damage index.

## 4. Conclusions

This work proposed an integrated biorefinery concept to produce value-added products (namely, xylitol, levulinic acid, succinic acid, guaiacol, and vanillin) from EFB. It formulated three cases of MOO: one for maximization of profit and minimization of GWP, another for maximization of profit and minimization of both FEDI and TDI, and the last one considering all objectives simultaneously, and solved them using surrogate models and a multi-objective genetic algorithm. The resulting Pareto-optimal solutions show the trade-off among the economic, environmental, and safety performances. One of these optimal solutions was selected using FAHP and TOPSIS. The preferred optimal solution revealed that the supply of EFB was at 100 ton/h, i.e., at the maximum supply of EFB possible to the plant. The optimal mass allocation of glucose was 10% to levulinic acid and the remaining to succinic acid, whereas 70% of lignin went to guaiacol and the rest to vanillin. The higher global demand of succinic acid led to more glucose allocation to it, while 10% for levulinic acid was for higher profit and lower GWP and TDI. Similarly, 70% of lignin went to producing more guaiacol than vanillin; 30% of lignin was used for vanillin production due to its better economics and lower environmental impact and TDI. All of xylose was allocated to produce xylitol since it had no other option. Succinic acid (22 ton/h), xylitol (18.8 ton/h), and guaiacol (5 ton/h) were the most-produced chemicals due to their higher global demand; others are levulinic acid (1.83 ton/h) and vanillin (0.55 ton/h). The optimal integrated biorefinery has an annual profit of $932 M USD/year, GWP of 284 tonCO$_2$-eq, FEDI of 595, and TDI score of 957. Further work on the utilization and optimization of other palm oil biomass to produce marketable bioenergy and biochemicals should be pursued to enhance the sustainability of palm oil industry. Also, cumulative energy cost should be included in MOO to enhance the sustainability of the integrated biorefinery.

**Supplementary Materials:** The following are available online at http://www.mdpi.com/2227-9717/8/7/868/s1, Figure S1: total global warming potential, Table S1: the estimation of capital investment, Table S2: the estimation of operating costs, Table S3: raw material and product selling price, Table S4: economic results for 75 ton/hour of dry empty fruit bunch, Table S5: inherent safety results summary, Table S6: decision maker attribute, Table S7:

TOPSIS key calculation information for profit, global warming potential, and fire explosion damage index, Table S8: TOPSIS key calculation information for profit, global warming potential, and toxicity damage index.

**Author Contributions:** Conceptualization, Z.A.P. and R.H.H.; resources, L.K.B. and R.H.H.; writing—original draft preparation, R.H.H.; writing—review and editing, R.H.H., M.R.B., N.A.H.N., M.D.H.W., Z.A.P., G.P.R. and B.A.; supervision, S.M.; funding acquisition, B.A. All authors have read and agreed to the published version of the manuscript.

**Funding:** This research and APC was funded by Yayasan Universiti Teknologi PETRONAS grant number: 015LC0-268.

**Acknowledgments:** The authors would like to thank Universiti Teknologi PETRONAS for providing financial assistance under YUTP grant (015LC0-268) and research facilities to conduct this research work.

**Conflicts of Interest:** The authors declare no conflict of interest.

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
