# Peer review of "Integrated Biorefinery of Empty Fruit Bunch from Palm Oil Industries to Produce Valuable Biochemicals"

_processes, doi:10.3390/pr8070868_

Round 1

Reviewer 1 Report

Dear Authors,

Everything looks all right, only some texts on Figs 2,6,7,9,10,11,14 are too small to read.

Best regards

the Reviewer

Reviewer 2 Report

Please to see enclosed review report (My Revision Report_processes-812991).

Reviewer 3 Report

Manuscript ID processes-812991entitled Integrated Biorefinery of Empty Fruit Bunch to Produce Biochemicals presents a very interesting and current issue related to optimization of production valuable bio-chemicals is seen as an economical and sustainable alternative to waste management in palm oil industries. In my opinion this is a very important topic due to the need to search for technologically, economically and ecologically justified systems for the production of biocomponents from broadly understood biomass. The Authors used many methods to evaluate and optimize the analyzed process. The manuscript contains extensive and very valuable material and can be published in the Processes journal after minor corrections and changes.

  1. The manuscript title should be extended and supplemented because the work concerns empty fruit bunch (EFB) utilization to produce valuable bio-chemicals is seen as an economical and sustainable alternative to waste management in palm oil industries. In my opinion, the title should be "Integrated Biorefinery of Empty Fruit Bunch from Palm Oil Industries to Produce Valuable Biochemicals"
  2. The Abstract section should not list all the techniques used to analyze and optimize the process. More space should be devoted to presenting the obtained results.
  3. The conclusions are too extensive. The authors once again repeat the purpose and methods of analysis used in them, namely: "An integrated biorefinery concept producing higher value-added products from palm oil biomass waste was proposed in this work. A MOO approach has been presented considering simultaneously three pillars of sustainability (economic, environment, and safety). The considered objectives are annual profit, global warming potential (GWP), and both fire and explosion damage index (FEDI) and toxicity damage index (TDI). Genetic algorithm based multi-objective algorithm (gamultiobj) was then employed for simultaneously optimizing these MOO problems. The obtained Pareto-optimal front provides decision-makers with many optimal solutions, where the trade-off is seen among the objective functions. To choose one of these optimal solutions, a combined approach of Fuzzy Analytical Hierarchy Process (FAHP) and Technique for Order Preference by Similarity to Ideal Solution (TOPSIS) was applied. " I believe that this fragment should be removed from the Conclusions section. More space should be devoted to the presentation of the results of the analyzes carried out.

I rate the manuscript very high. The Authors have put a lot of work into carrying out a multivariate analysis leading to process optimization. In my opinion, the manuscript should be published in Prcesses.
